# A Novel Phenazine Analog, CPUL1, Suppresses Autophagic Flux and Proliferation in Hepatocellular Carcinoma: Insight from Integrated Transcriptomic and Metabolomic Analysis

**DOI:** 10.3390/cancers15051607

**Published:** 2023-03-05

**Authors:** Jiaqin Chen, Dong Feng, Yuanyuan Lu, Yanjun Zhang, Hanxiang Jiang, Man Yuan, Yifan Xu, Jianjun Zou, Yubing Zhu, Jingjing Zhang, Chun Ge, Ying Wang

**Affiliations:** 1School of Life Science and Technology, China Pharmaceutical University, Nanjing 211198, China; 2Nanjing Southern Pharmaceutical Technology Co., Ltd., Nanjing 211100, China; 3Department of Clinical Pharmacy, School of Basic Medicine & Clinical Pharmacy, China Pharmaceutical University, Nanjing 210009, China; 4Department of Pharmacy, Nanjing First Hospital, Nanjing Medical University, Nanjing 210006, China; 5Department of Clinical Pharmacology, Nanjing First Hospital, Nanjing Medical University, Nanjing 210006, China

**Keywords:** CPUL1, hepatocellular carcinoma, transcriptomics, metabolomics, autophagy

## Abstract

**Simple Summary:**

CPUL1 exhibits antitumor properties against hepatocellular carcinoma (HCC), although the underlying mechanisms remain unclear. Our study provides a comprehensive overview of the properties and molecular mechanisms of CPUL1 anti-HCC using transcriptomics and metabolomics and highlights the significance of progressive metabolic failure. This may be partially attributable to autophagy blockage, which is presumed to contribute to nutrient deprivation and increased cell susceptibility to stress. Therefore, the attractive properties of CPUL1 may endow this compound with the potential of becoming a promising anti-HCC agent.

**Abstract:**

Background: CPUL1, a phenazine analog, has demonstrated potent antitumor properties against hepatocellular carcinoma (HCC) and indicates a promising prospect in pharmaceutical development. However, the underlying mechanisms remain largely obscure. Methods: Multiple HCC cell lines were used to investigate the in vitro effects of CPUL1. The antineoplastic properties of CPUL1 were assessed in vivo by establishing a xenograft nude mice model. After that, metabolomics, transcriptomics, and bioinformatics were integrated to elucidate the mechanisms underlying the therapeutic efficacy of CPUL1, highlighting an unanticipated involvement of autophagy dysregulation. Results: CPUL1 suppressed HCC cell proliferation in vitro and in vivo, thereby endorsing the potential as a leading agent for HCC therapy. Integrative omics characterized a deteriorating scenario of metabolic debilitation with CPUL1, presenting an issue in the autophagy contribution of autophagy. Subsequent observations indicated that CPUL1 treatment could impede autophagic flow by suppressing autophagosome degradation rather than its formation, which supposedly exacerbated cellular damage triggered by metabolic impairment. Moreover, the observed late autophagosome degradation may be attributed to lysosome dysfunction, which is essential for the final stage of autophagy and cargo disposal. Conclusions: Our study comprehensively profiled the anti-hepatoma characteristics and molecular mechanisms of CPUL1, highlighting the implications of progressive metabolic failure. This could partially be ascribed to autophagy blockage, which supposedly conveyed nutritional deprivation and intensified cellular vulnerability to stress.

## 1. Introduction

As a principal pathological type of liver cancer, hepatocellular carcinoma (HCC) is characterized by an increasing incidence and subsequent mortality [1]. High recurrence, phenotypic and genetic heterogeneity, and metastasis confer poor prognosis and disappointing survival rate in HCC, which has posed a substantial health care burden worldwide [2]. Currently, chemotherapy is one of the systemic therapies for HCC in the clinic, with sorafenib and lenvatinib being the most effective single-drug regimens [3]. In contrast, the efficacy of sorafenib, the canonic first-line treatment for advanced HCC, remains moderate, with a median survival duration of less than one year and a tumor response rate of less than 5% [4]. There remains a critical and unmet demand for innovative and effective therapeutic options against HCC.

Chemically, phenazines are a class of nitrogen-containing aromatic compounds of natural or synthetic origin. Their redox properties confer broad biological functions including antimicrobial, insecticidal, antimalarial, antitumor, and antiplatelet properties [5,6,7,8,9]. Our previous research focused on the design and synthesis of phenazine derivatives for chemotherapeutic screening and optimization, which led to the discovery of CPUL1, a promising lead compound. The compound demonstrated potent alleviative effects against HCC, represented by the apoptosis of HepG2 cells in vitro and the regression of H22 xenograft tumor in vivo [10]. Earlier research has observed that CPUL1 treatment could suppress thioredoxin reductase I (TrxR1), a central component in the thioredoxin system that maintains cellular redox homeostasis and protects against oxidative stress [11]. Therefore, excessive reactive oxygen species (ROS) are generated and accumulated, contributing to lipid peroxidation, DNA damage, and cell apoptosis, as confirmed by cytology and morphology [10]. Nonetheless, the precise functions and biological mechanisms of CPUL1 in HCC remain unclear, which presumably involves a complicated pathophysiological process and necessitates further research.

The advent of omics techniques, and their extensive implementation, provides an operational platform for profiling intricate biological systems [12]. Transcriptomics and metabolomics reflect the systematic alterations in the genotype and phenotype and demonstrate comprehensive information comprising genetic regulation, protein synthesis, metabolic pathways, and cellular functions [13]. Moreover, multi-omics data integration offers a more reliable methodology and a holistic perspective for illuminating the complex biological scenario [14], presumably facilitating accurate and rational insights for research practice, especially for complex diseases and therapeutic regimens.

In this regard, we sequentially evaluated the therapeutic effects of CPUL1 in human HCC cells BEL-7402 in vitro and in vivo and integrated the transcriptomics, metabolomics, and bioinformatics analyses to illuminate the underlying multi-threading mechanisms. The investigations present a panorama of genetic and metabolic profiles altered by CPUL1, with a particular focus on autophagy restriction, which not only substantiates the translational potentials of this lead compound, but also sheds some light on the chemotherapeutic interventions toward HCC.

## 2. Materials and Methods

### 2.1. Cell Culture and Reagents

The human HCC cell lines HepG2, BEL-7402 (KeyGen Biotech, Nanjing, China), and HUH-7 (FuHeng Cell Center, Shanghai, China) were cultured in DMEM or RPMI-1640 (KeyGEN Biotech, Nanjing, China) supplemented with 10% fetal bovine serum (Gibco/BRL, New York, NY, USA), penicillin (80 units/mL), and streptomycin (80 mg/L) at 37 °C under 5% CO_2_. Compound CPUL1 was synthesized as described previously (China Pharmaceutical University, Nanjing, China), with a purity of 98% determined by RP-HPLC. Other reagents and solvents purchased were of analytical grade and obtained from commercial companies. Chloroquine (CQ) was procured from KeyGEN Biotech Co., Ltd. (Nanjing, China), and 3-methyladenine (3-MA) was obtained from MedChemExpress (Middlesex, NJ, USA).

### 2.2. Cell Viability Assay

According to the manufacturer’s instructions, the cell inhibition rate was determined using the Cell Counting Kit-8 (CCK-8; Beyotime, Shanghai, China). Briefly, 4 × 10^4^ cells were seeded in 96-well plates and treated with various concentrations of CPUL1 for 48 h. The optical density of each sample was measured at 450 nm on an automatic plate reader (ALLSHENG, Hangzhou, China). The half-maximal inhibitory concentration (IC_50_) of CPUL1 for different cells was calculated according to the standard curve.

### 2.3. Cell Colony Formation Assay 

A colony formation assay was used to evaluate the cell proliferation capacity. Approximately 1 × 10^3^ cells were cultured in 6-well plates and treated with CPUL1 at 0, 1, 2, and 5 μM. After 14 days, the cells were fixed with 4% paraformaldehyde for 20 min, stained with crystal violet (Beyotime, Shanghai, China) for 10 min, and washed thrice with phosphate-buffered saline (PBS). To estimate the number of colonies formed, crystal violet was dissolved in 95% ethanol, transferred to a 96-well plate, and absorbance was measured at 590 nm.

### 2.4. Subcellular Localization of CPUL1

The subcellular distribution of CPUL1 was investigated in the BEL-7402 cells (≈1 × 10^7^). The adherent cells were treated with CPUL1 at 8 μM for 6 h, 24 h, and 48 h, respectively. Following treatment, the cells were collected by centrifugation. After lysis, the mitochondria, nucleus, and cytosolic fractions were isolated using a mitochondria/nuclei isolation kit (KGA828, KeyGEN Biotechnology, Nanjing, China) according to the manufacturer’s instructions. LC-MS then determined the concentrations of CPUL1 in different organelles with a 4000 Q Trap (Applied Biosystem, Foster City, CA, USA).

### 2.5. Antitumor Activity In Vivo

The antitumor activity of CPUL1 was evaluated in vivo in a BEL-7402 xenograft model. The male immunodeficient nude BALB/c-nu-nu mice (4~5 weeks, 18–20 g) were purchased from Lingchang Biotechnology Co., Ltd. (Shanghai, China). Mice were maintained in SPF facilities under a controlled environment (22 °C–24 °C, 50–60% humidity, 12 h light/12 h dark cycle) with ad libitum access to standard laboratory food and water. All animal care and experimental procedures were conducted in accordance with the Institutional Animal Care and Use guidelines (IACU, China Pharmaceutical University) and the 3R principles.

To establish a xenograft model, the BEL-7402 cells (1 × 10^7^) were subcutaneously injected into mice at the right axillae. When the tumor volume reached approximately 90 mm^3^, the mice were randomly divided into six groups (*n* ≥ 6): control, sorafenib (20 mg/kg); cyclophosphamide (CTX, 20 mg/kg); CPUL1(10, 20, and 40 mg/kg). Before treatment, the test drugs were dissolved in dimethyl sulfoxide (DMSO) and diluted by saline containing Tween-80 [final vehicle: DMSO/Tween/0.9% NaCl (1.5/1.5/97, *v*/*v*)]. All groups were administered with vehicle (control), positive agents (sorafenib or cyclophosphamide), or CPUL1 at a dosing volume of 10 mL/kg. Administrations were performed by intragastric gavage (i.g., sorafenib and cyclophosphamide groups) or tail vein injection (control and CPUL1 groups) once a day continuously for a week. During therapy, the tumor volume (length × width^2^/2; mm^3^) and body weight of the mice were measured every other day. At the end of the experiment, mice were sacrificed, and tumors were isolated and weighed.

### 2.6. RNA Sequencing (RNA-Seq) Analysis

Sample preparation: For RNA-Seq analysis, the BEL-7402 cells treated with CPUL1 at 0 μM (DMSO) and 8 μM for 6 h and 48 h were used. In each group, the samples were tripled. The total RNA was extracted using TRIzol reagent (Invitrogen, Carlsbad, CA, USA). RNA quantity and quality were determined using a Nanodrop spectrophotometer (Thermo Fisher Scientific, Fremont, CA, USA) and an Agilent 2100 instrument (Agilent Technologies, Santa Clara, CA, USA), respectively. The samples with an RNA integrity number of >7 and OD_260/280_ of >1.8 were used for library construction.

Library construction and sequencing: The cDNA library was constructed using 2 μg of RNA with a NEBNext Ultra^TM^ RNA Library Prep Kit for Illumina (NEB, Ipswich, MA, USA) according to the manufacturer’s instructions. All libraries were loaded onto the Illumina NovaSeq 6000 platform (Illumina, San Diego, CA, USA), followed by 2 × 150 bp paired-end read sequencing. 

Data processing and functional analysis: The raw reads in fastq format were processed using in-house Perl scripts. Clean reads were obtained by removing low-quality reads and those with adapters or poly-N sequences. Q20, Q30, GC content, and sequence duplication level were calculated for the clean data. High-quality sequences were aligned to the reference genome of BEL-7402 (ENSEMBL, Homo sapiens. GRCh38.101) using Hisat2 (v2.0.1) with the default parameters. Cufflinks, HTSeq (v. 0.5.4 p3), and DESeq (v. 1.10.1) were used for assembly, mRNA expression level evaluation, and differentially expressed gene (DEG) identification, respectively. Genes with Log_2_|fold change (FC)| >1 and *Q*-value ≤ 0.05 were considered DEGs. 

Functional analysis: All DEGs were implemented into GOSeq (v. 1.34.1) to identify Gene Ontology (GO) terms that annotate a list of enriched genes with a *p*-value ≤ 0.05. Meanwhile, DEGs were also subjected to the Kyoto Encyclopedia of Genes and Genomes (KEGG) database for pathway enrichment with a *Q*-value ≤ 0.05. 

Construction of protein–protein interaction network and screening of hub genes: The protein–protein interaction (PPI) network was constructed among the screened DEGs by the online database STRING [15]. The retrieved interaction networks were visualized by Cytoscape (v3.6.1) and calculated in the cytoHubba control panel for hub node prediction. The genes with the top ten values ranked by the Maximal Clique Centrality (MCC) algorithm were considered hub genes [16].

### 2.7. Metabolomics Analysis 

Collection and preparation of samples: The samples used for RNA-Seq analysis were also used for the LC-QTOF/MS analysis. The cells were washed with PBS, freezing-thawed three times, and normalized by the BCA assay (KeyGEN Biotech, Nanjing, China). The cell pellets were resuspended in methanol containing the internal standard (L-Tryptophan(^13^C_11_)/L-Phenylalanine(^13^C_6_), followed by shaking for 5 min to lyse the cells and release cellular metabolites. After centrifugation at 18,000 rpm and 4 °C for 10 min, the supernatant was obtained and evaporated. The residues were re-dissolved in 100 μL ice-cold methanol/water (1:1, *v*/*v*), and the supernatant was collected for liquid chromatography quadrupole time-of-flight mass spectrometry (LC-QTOF-MS) analysis. 

Acquisition of LC-QTOF-MS data: The relative amounts of each metabolite were obtained by integrating peaks detected on a Sciex TripleTOF 5600 (AB SCIEX, Foster City, CA, USA). Chromatographic separation was performed on a Waters HSS T3 1.8 μm, 2.1 × 100 mm column. The mobile phase for chromatographic elution was an online mixing of phase A and phase B, where phase A was water containing 0.1% formic acid and phase B was acetonitrile. Compounds were identified by matching chromatographic retention time and mass spectral fragmentation signatures with the reference library data created from authentic standards. Each treatment group comprised eight biological replicates.

LC-QTOF/MS data processing: The raw data were processed with MS-DIAL (v. 4.70). Statistical data analysis was performed using SIMCA-P 14.1. Both univariate and multivariate data analysis were hired to extract information from the metabolomics datasets. A multivariate statistical method was then conducted to classify samples with optimal variables, which was cross-validated by principal component analysis (PCA), partial least squares discrimination analysis (PLS-DA) [17], and orthogonal projections to latent structures-discriminant analysis (OPLS-DA) [18]. The differential abundance metabolites (DAMs) were screened as significantly affected metabolites using multiple statistical approaches. The screening criteria were as follows: the “variable importance in projection” (VIP) scores of the OPLS-DA model greater than 1, the *p*-values from a *t*-test < 0.05, and the fold change values greater than 2 (*p* < 0.05, OPLS-DA-VIP > 1.0 and |FC| > 2) [19]. Then, the DAMs were mapped in the KEGG database, whereas pathway enrichment analysis was performed using the built-in function available in the MetaboAnalyst 5.0 server (www.metaboanalyst.ca accessed on 1 November 2022) [20].

### 2.8. Integrated Analysis of Transcriptomics and Metabolomics

Based on transcriptome and metabolome analysis data, we obtained differential genes and differential metabolites. Pathway enrichment analysis was performed using the MetaboAnalyst 5.0 server (www.metaboanalyst.ca accessed on 1 November 2022) to describe in detail the relationship between gene regulation and metabolic changes, with a *p*-value < 0.05 considered to be significant.

### 2.9. Western Blotting Analysis

The cells were lysed with RIPA buffer (Beyotime, Shanghai, China) containing a mixture of PMSF (Beyotime, China). Then, the lysates were cleared by centrifugation, and the concentrations of proteins were measured by the BCA Protein Assay Kit (Beyotime, China). Each protein sample (20 µg) was separated on SDS-PAGE gels and transferred to PVDF membranes (Millipore, Milford, DE, USA). The membranes were placed in 5% skim milk for 1 h at room temperature and incubated with antibodies of AMPK (WL02254; 1:1000; Wanleibio Co., Ltd., Shanghai, China), p-AMPK (WL05103; 1:1000; Wanleibio Co., Ltd.), ATG5 (WL02411; 1:1000; Wanleibio Co., Ltd.), and ATG12 (WL03144; 1:1000; Wanleibio Co., Ltd.), LC3 (14600-1-AP; 1:1000; Proteintech Group, Inc., Wuhan, China), p62 (66184-1-Ig; 1:1000; Proteintech Group, Inc.), mTOR (66888-1-Ig; 1:1000; Proteintech Group, Inc.), p-mTOR (67778-1-Ig; 1:1000; Proteintech Group, Inc.), and Beclin-1(66665-1-Ig; 1:1000; Proteintech Group, Inc.), LAMP1(67300-1-Ig; 1:20,000; Proteintech Group, Inc.), LAMP2(WL02761; 1:500; Wanleibio Co., Ltd.), RAb7(ET1611-96; 1:1000; HUABIO, Woburn, MA, USA), and GAPDH (AF7021; 1:1000; Affinity, San Francisco, CA, USA) at 4 °C overnight. Then, the samples were incubated with the secondary goat anti-rabbit IgG-HRP (BL001A, Biosharp, Tallinn, Estonia), conjugated with horseradish peroxidase for 1 h at room temperature. The bands were visualized using an image analysis system (Tanon, Shanghai, China).

### 2.10. Transmission Electron Microscopy (TEM) 

Autophagosomes were observed using TEM. The cells were fixed in 2.5% glutaraldehyde (pH 7.3–7.4) at 4 °C overnight and treated with 1% osmium tetroxide for 2 h. Then, the samples were dehydrated in ethanol (30%, 50%, 70%, 80%, 90%, and 95%) and pure acetone, embedded, cut into 90 nm sections, and stained with 3% uranyl acetate and lead citrate. TEM visualizations were performed using a Hitachi H-9000 transmission electron microscope at 300 kV, and images were captured using a slow-scan CCD camera.

### 2.11. Immunofluorescence 

The cells were fixed with 4% paraformaldehyde for 20 min and permeabilized with 0.1% Triton X-100 for 5 min. The cells were incubated overnight with LC3 (14600-1-AP; 1:400; Proteintech Group, Inc.), p62 (66184-1-Ig; 1:800; Proteintech Group, Inc.), or LAMP1 (67300-1-Ig; 1:800; Proteintech Group, Inc.), antibodies at 4 °C on a horizontal shaker. Then, the cells were incubated with a Cy3-labeled goat anti-rabbit IgG (H+L) secondary antibody (Beyotime, China) or Alexa Fluor 647-labeled goat anti-rabbit IgG (H+L) secondary antibody (Beyotime, China) for 1 h at room temperature, washed with PBS, and stained with DAPI (Beyotime, China) for 5 min. The photographs were taken under laser scanning confocal microscopy (CLSM, LSM800, Zeiss, Oberkochen, Germany).

### 2.12. Lyso-Tracker Red Staining

Following CPUL1 administration or not, cells were stained with Lyso-Tracker Red (50 nM), a cell-permeable red-fluorescent dye for lysosomes, for 1 h at 37 °C and then counterstained with Hoechst 33342 (1 μM) for 10 min at room temperature in the dark. After washing by PBS, the cells were observed under a fluorescence microscope [21].

### 2.13. Statistical Analysis

All data are expressed as the mean ± standard deviation (SD) of at least three independent experiments, except for eight biological replicates of metabolomics. Two-tailed Student’s *t*-tests and one-way analysis of variance were employed for statistical analysis. Differences were considered significant at * *p* < 0.05, ** *p* < 0.01, *** *p* < 0.001. Statistical data analysis was performed using GraphPad Prism 8 (GraphPad Software Inc., San Diego, CA, USA).

## 3. Results

### 3.1. CPUL1 Exerted Antitumor Effects against HCC In Vitro 

Three HCC cell lines (HUH-7, HepG2, and BEL-7402) were used to investigate the in vitro effects of CPUL1 (Figure 1A). The cell viabilities of HCC cells were assessed by the CCK-8 assay, following 48 h treatment of CPUL1 at the indicated concentrations, which consistently revealed a dose-dependent inhibition pattern. The cytotoxic potencies were evident in different HCC cells, represented by the approximately comparable IC_50_ values, respectively, at 4.39 μM (HUH-7; Figure 1B), 7.55 μM (HepG2; Figure 1C), and 6.86 μM (BEL-7402; Figure 1D). Hence, BEL-7402 cells were used in subsequent experiments.

Morphological changes were then observed in the BEL-7402 cells treated with 8 μM CPUL1 at different time points (6 h and 48 h), which induced rounding, shrinking, and crushing, in a time-dependent manner (Figure 1E). The colony formation assay further confirmed the antiproliferative effect of CPUL1 as the colony-forming efficiency was significantly decreased, close to a depletion level at a 5 μM concentration (*p* < 0.001; Figure 1F). The suborganelle localization of CPUL1 in the HCC cells was identified by LC-MS-based concentration measurements, which showed that the compound was predominantly distributed in the cytoplasm and mitochondria. In contrast to the relatively steady concentration in the cytoplasm, significant accumulations occurred gradually within the mitochondria and nuclei (Figure 1G), presumably underpinning the mitochondrial dysfunction and DNA damage previously observed [10]. 

### 3.2. CPUL1 Exerted Antitumor Effects against HCC In Vivo

Correspondingly, the BEL-7402 xenograft model was constructed to further evaluate the therapeutic potential of CPUL1 in vivo (Figure 1H). As expected, CPUL1 significantly inhibited the tumor growth at various concentrations, with potencies comparable or even superior to those of the positive controls (sorafenib and CTX; 20 mg/kg) at higher dosages (40 and 20 mg/kg), in terms of tumor volume and weight (Figure 1I,K). In contrast, the body weight of mice increased slightly during therapy, without statistically significant differences among all groups (Figure 1J). Hematoxylin and eosin staining showed no tumor cell necrosis in the model control group and cyclophosphamide group (Figure 1L). In contrast, both the CPUL1 administration group and sorafenib group showed tumor tissue necrosis to a certain extent, and the necrotic area of the tumor tissue in the CPUL1 administration group increased with the increase in the administration concentration, indicating that CPUL1 had obvious toxic effects on tumor cells and tissues in an obvious dose-dependent manner. In vitro and in vivo investigations consistently demonstrated the promising potential of CPUL1 for HCC therapy, justifying further studies of the detailed mechanisms.

### 3.3. Transcriptomics Analysis on CPUL1-Treated BEL-7402 Cells

RNA-Seq analysis was performed to profile the transcriptomic changes in BEL-7402 cells treated with 8 μM of CPUL1 for 6 h and 48 h. A total of 40.44 Mb reads were generated in the control group, 42.80 Mb reads in the 6 h-treated group, and 46.71 Mb reads in the 48 h-treated group. After screening, 36.30–49.24 Mb clean reads remained (Appendix A). Sample Q30 was >92.22%, and the average GC content was 52.60–53.60%. 

By pairwise comparisons, a total of 989 (598 downregulated and 391 upregulated) DEGs were identified from the 6 h_treated_vs._Control groups; and 1207 DEGs including 333 upregulated and 874 downregulated genes were identified in the 48 h_treated_vs._Control comparisons. All DEGs from both groups are cataloged in Appendix A, with detailed information sketched in the volcano plots (Figure 2A). As the CPUL1 exposure was prolonged, the number of downregulated transcripts increased from 598 to 874, whereas the number of upregulated transcripts decreased from 391 to 333. Moreover, 416 DEGs maintained a consistent pattern at the indicated time points, either descending (335 DAGs) or ascending (81 DEGs) in comparison with the control (Figure 2B). The transcriptional alterations were also represented in the cluster heat map, potentially implying an intricacy and multiplicity of genetic interactions and functional integrations (Figure 2C).

#### 3.3.1. GO and KEGG Enrichment Analysis 

Transcriptome profiles were integrally analyzed on DEGs by pathway-enrichment analysis using the GO and KEGG databases (Figure 3). The GO term analysis on the DEGs from 6h_vs._Control highlighted the significantly enriched functions involving “sequestering of TGF-beta in the extracellular matrix”, “collagen type IV trimer”, and “insulin-like growth factor-activated receptor activity” (Figure 3B). In contrast, the 48 h-treated DEGs were mainly enriched in “cell–cell adhesion mediated by integrin”, “laminin-1 complex”, and “ATP-dependent 3′-5′ DNA helicase activity” (Figure 3C).

Regarding functional annotation, DEGs in the 6 h_vs._Control and 48 h_vs._Control comparisons were enriched to the 290 and 301 KEGG pathways, with an overlap of 277 identical pathways (Figure 3D). Even among the top 30 significantly altered pathways, the similarity in functional mapping was also apparent, with the most significantly enriched pathways converged in “valine, leucine, and isoleucine biosynthesis”, “MAPK signaling pathway”, “PI3K-Akt signaling pathway”, “mitophagy-animal”, and “autophagy-animal” (Figure 3E,F). Notably, “biosynthesis of valine, leucine and isoleucine” was the most enriched pathway in both groups, which assumably implied a substantial perturbation in metabolism due to CPUL1 treatment.

#### 3.3.2. PPI Network Analysis

Using STRING and Cytoscape, the DEGs from pairwise comparison were assembled into biological networks at a high confidence cutoff (interaction score ≥ 0.9) to investigate the major events from a complex interactome aspect. As shown in Figure 4A, the predictive network among DEGs from 6 h_vs._Control, consisting of 86 nodes and 313 edges, clustered into biological schemes also concerning “arginine and proline metabolism”, “inositol phosphate metabolism”, “WNT signaling pathway”, TGF-beta signaling pathway”, and “ECM–receptor interaction”. Comparatively, DEGs in the 48 h_vs._Control group were enriched in “WNT signaling pathway”, “TGF-beta signaling pathway”, and “ECM–receptor interaction”, which constructed a complex network by 125 nodes and 268 edges (Figure 4B).

Hub genes, defined as nodes with high degrees of connectivity and centrality within a module, confer heuristic insights into the essential mechanisms beneath the physiological and pathological processes [22]. According to the MCC sores, the hub genes were screened by group, and are shown in Appendix A. Specifically, hub genes in the 6 h-treated group included ITGB1, EGFR, LAMB1, LAMC1, HSPG2, COL4A1, COL4A2, ITGA1, ITGA5, and ITGA6, which appeared to be implicated in carcinogenesis [23]. As for the 48 h-treated group, a hub gene panel was identified, consisting of ITGA1, ITGA2, ITGB1, EP300, CREBBP, BRCA1, LAMB3, LAMC1, LAMA3, and LAMB1. 

Functional analysis was performed on the hub gene collection utilizing the DAVID database for GO terminology and KEGG pathway enrichment. GO analysis indicated that hub genes were enriched in the canonical cellular processes associated with tumor progression (e.g., “extracellular matrix organization”, “cell adhesion”, and “cell migration”). Consonantly, the hub genes were also overrepresented in the KEGG pathways involving cancer occurrence and progression including “ECM-receptor interaction”, “focal adhesion”, and “PI3K-Akt signaling” (Appendix A) [24]. Intriguingly, for either time point of CPUL1 treatment, the panel of hub genes or annotated pathways exhibited subtle differences, albeit within a generally consistent perspective, which to some extent, presumably implied a temporal pattern of the signaling network.

### 3.4. Non-Targeted Metabolomics Revealed Metabolic Perturbation Networks by CPUL1

Both the functional annotations and pathway enrichments of DEGs proposed a correlative and progressive change in the signaling panels during CPUL1 exposure, orchestrating the shift from metabolic regulation to cell fate and highlighting a chronological accumulation of metabolic perturbations. Hence, untargeted global metabolomics analysis was performed to profile the metabolites altered by compound therapy. 

The initial LC-QTOF/MS data were applied to the multivariate principal component analysis to determine cluster separations among the experimental groups (PCA and PLS-DA plots shown in Appendix A). Pairwise comparison of the metabolite profiles respectively identified 229 (59 upregulated and 170 downregulated) and 222 (83 upregulated and 139 downregulated) putative DAMs from the 6 h_vs._Control or 48 h_vs._Control group (Appendix A). 

Notably, the DAMs exhibited temporal discrepancies in panel composition or abundance level (Figure 5A). As shown in Figure 5B,C, for a range of putative DAMs, the substantial abundances were exclusively changed at a particular time point (6 h or 48 h) after treatment: formononetin, glucose 6-phosphate, phosphoenolpyruvic acid (PEP), were exclusively decreased at the late-stage (48 h), with FC values of 0.05, 0.11, and 0.15, respectively; however, asperuloside was explicitly abundant in 6 h (FC value ≈ 8.25). Interestingly, α-ketoglutarate (α-KG) and oleic acid evidenced the opposite dynamic at separate time phases, with their abundances declining (6 h) or ascending (48 h) in comparison with the control group. For most DAMs, the variation tendency was undoubtedly consistent, albeit the magnitude fluctuated over time.

Correspondingly, further functional analysis on DAMs at different time phases also demonstrated a real discernible distinction in biochemical pathways, presumably proposing the difference in global metabolic status induced by CPUL1 treatment maintenance. As KEGG enrichment profiling revealed, DAMs identified in the 6 h_vs._Control groups mainly involved in “arginine biosynthesis”, “D-glutamine and D-glutamate metabolism”, “purine metabolism”, “alanine, aspartate and glutamate metabolism”, “linoleic acid metabolism”, “pantothenate and CoA biosynthesis”, and “pyrimidine metabolism” (Figure 5D); in contrast, “purine metabolism”, “pyrimidine metabolism”, “glutathione metabolism”, “linoleic acid metabolism”, “pentose and glucuronate interconversions”, “pentose phosphate pathway”, and “arginine biosynthesis” were enriched by DAMs from the 48 h_vs._Control group (Figure 5E).

### 3.5. Integrative Omics Analysis Highlighted a Temporal Variegation of Metabolic Responses 

The DAGs and DAMs were merged to construct the transcript–metabolite interaction network to depict the biological responses induced by CPUL1 therapy at the indicated times. During the integration of the transcriptomics and metabolomics data of the initial therapy (6 h), seven pathways were significantly enriched (*p* < 0.05) including “arginine biosynthesis”, “alanine, aspartate and glutamate metabolism”, “glycerolipid metabolism”, “histidine metabolism”, “purine metabolism”, “arginine and proline metabolism”, and “inositol phosphate metabolism” (Figure 6A); but four pathways, “purine metabolism”, “mucin type O-glycan biosynthesis”, “lysine degradation”, and “pyrimidine metabolism” were evidently perturbed during the late phase (48 h; Figure 6B). Integrated analysis also confirmed a temporal heterogeneity of the global cellular response to CPUL1 treatment, either transcriptional regulations or metabolic reactions (Figure 6C). As for the integration pathways in the respective dosing durations, the DAMs were significantly downregulated, except for guanosine diphosphate (GDP) and 3′,5′-cyclic GMP (cGMP), which demonstrated a substantial dysregulation in multiple biological processes, despite the transcriptional upregulation of a proportion of genes (e.g., GPT, TKFC, HDDC3, NME1, PYCR3, PIP5KL1, PLCB2, and GLYCTK) (Appendix A). 

### 3.6. CPUL1 Therapy Exacerbated Metabolic Dysregulation and Cellular Malnutrition in HCCs

Given the prominence of amino acid metabolism in tumorigenesis and metastasis [25], the early restriction (6 h) of amino acids could presumably hamper metabolic flexibility and induce extensive dysfunctions. Hence, based on the relative quantification of metabolomics, the cellular-scale nutrients and metabolic intermediates were assessed, which profiled a scenario of metabolic disorder and nutritional restriction characterized by the depletion of multitudinous metabolites (Figure 7). For instance, the relative abundances of several glycolysis intermediates including glucose 6-phosphate (FC ≈ 0.11), 3-phospho-D-glycerate (FC ≈ 0.20), and PEP (FC ≈ 0.15) were continuously decreased under 48 h-exposure of CPUL1, which might represent a moderating glycolysis rate and is consistent with the lower ATP level at 48 h (FC ≈ 0.34) (Figure 5C). The comparable pattern also recurred in diverse biological processes such as the metabolism of nucleotides, arginine, and proline, histidine, glutathione biosynthesis, and energy production (Figure 7), collectively indicating a deteriorative status of metabolic dysfunction and nutritional deficiency. 

### 3.7. The Blockage of Autophagic Flux by CPUL1 Might Implicate Metabolism Perturbations

Since CPUL1 was presumed as a cytotoxic agent, this raised a question as to what mechanisms would be underlying the metabolic disruption and malnutrition. Hence, we systemically examined the temporal dynamics of biological responses in the context of transcriptomic and biochemical modifications, which highlighted the putative involvement of autophagy, as it is not only a process tightly interconnected with nutrient status or metabolic stress, but also one of the enriched reactions by any approaches of data analysis in this study (Figure 3, Figure 5, Figure 6 and Figure 8A). 

Hence, genes associated with autophagy were screened from the transcriptomic data of different CPUL1-exposing times. As indicated in Figure 8A, many autophagy-related and regulatory genes were transcriptionally downregulated including AMPK, AKT3, PERK, and ATGs. The expression levels of the ATG4A, ATG9A, ATG12, and ATG5 genes in the autophagy pathway were verified by qPCR (Figure 8B). The results showed that the expression levels of the above genes were downregulated, which was consistent with the transcriptomic results. However, as autophagy is a highly dynamic and complex process regulated by multiple steps [26], it was challenging to predict autophagy flux or the consequence altered by those genes. The expressions of several hallmark autophagy proteins (e.g., AMPK, LC3, mTOR, and ATG) were assessed to speculate the autophagic status in HCC cells. The results suggest that CPUL1 dramatically inhibited the pan-expression of AMPK and mTOR in a time-dependent manner. However, the relative activation appeared in the opposite pattern, represented by the descending ratio of p-AMPK/AMPK and the ascending ratio of p-mTOR/mTOR, which presumably indicated the repression of autophagy. In addition, the expression of downstream protein C/EBP-β was downregulated, which was consistent with the expression trend of p-mTOR/mTOR and p-AMPK/AMPK. There was a decline in Beclin1, ATG5, and ATG12, the critical mediators of autophagy initiation and autophagosome formation [27]; however, the conversion of LC3-I to LC3-II was surprisingly increased, which often reflects autophagosome biogenesis. Subsequently, the expression level of p62, an indicator of autophagic flux, was assessed by Western blotting and QPCR, for which the significant intracellular accumulation was observed to imply the restraint of autophagosome degradation [28] (Figure 8D,E). The simultaneous increase in LC3-II and p62 indicated the initiation of autophagosome assembly, but the suppression of autolysosome degradation, which allegedly connotes an interception of autophagy flux [29]. Accumulation of autophagosomes was then confirmed using TEM micromorphological observations. The results show that double-membrane vesicles were prominently deposited in the cytoplasm following 6 h or 48 h of CPUL1 treatment, even though these structures were supposed to be transient and sporadic during the autophagy process (Figure 8C). 

To exclude the possibility that the observed autophagy disruption induced by CPUL1 was stochastic or cell-specific, we performed supplementary experiments on additional HCC lines, HepG2, and HUH-7. Western blotting and immunofluorescence analyses demonstrated elevated expressions of LC3-II and p62 after 6 h and 48 h of exposure to CPUL1 (particularly after 48 h) in the HepG2 and HUH-7 cells (Appendix A), consequently indicating the suppression of autophagy (Figure 8F,H), which were consistent with those obtained from the BEL-7402 cells. In light of the dynamic nature of the autophagic process, especially in response to nutrient deprivation (as occurs with CPUL1 treatment), it is crucial to investigate the temporal effect of CPUL1 on autophagy in a compact frame (i.e., 1 h, 2 h, and 4 h) to obtain insights into its underlying mechanisms. As revealed in Figure 8G, the protein expression levels of LC3 and p62 remained substantially unaltered following short-term exposure to CPUL1 in the investigated cell lines. This was corroborated by immunofluorescence assays (Figure 8I) with comparable protein depositions of LC3 and p62, regardless of 2-h treatment of CPUL1 or not (Appendix A). These observations indicated that brief exposure to CPUL1 is unlikely to induce an immediate suppression on autophagy, but rather a cumulative effect, possibly due to its gradual and gentle intracellular accumulation, as indicated by the pharmacokinetic profiles [30].

The mechanism underlying the effect of CPUL1 on autophagy was subsequently investigated in BEL-7402 cells by measuring LC3-II turnover and p62 degradation, in combination with an autophagy inhibitor (CQ or 3-MA), to further validate the CPUL1 impairment on autophagic dynamics. Both immunoblotting and immunofluorescence staining (Figure 9A–C) revealed the considerable accumulations of LC3-II and p62 caused by CPUL1 treatment (Appendix A), comparable to that induced by CQ, a late-stage inhibitor by interrupting autophagosomal degradation [31]. Moreover, the co-incubation of CQ and CPUL1 slightly but significantly potentiated the trend, possibly implicating a parallel mode of action. In contrast, 3-MA, a typical early autophagy blocker [32], specifically increased the expression of p62 and LC3 proteins. Simultaneous treatment with CPUL1 partially intensified the 3-MA-induced effects, as demonstrated by a considerable augmentation in p62 abundance (Figure 9C). The aggregate of the above results suggest that the cellular autophagy flux in BEL-7402 was potently suppressed by CPUL1, presumably through an intervention on autophagosomal degradation in a manner analogous to CQ, which could conceivably exacerbate the metabolic disturbances and nutritional deficiencies.

The fusion of autophagosomes with lysosomes is a crucial step in autophagy, facilitating the delivery of autophagosome cargo for degradation by the lysosome [33]. To investigate the impact of CPUL1 on lysosome function, endogenous LC3 colocalization with LAMP1, a well-established lysosomal marker [34], was examined using immunofluorescence. Results showed that CPUL1 treatment decreased LAMP1 distribution as well as LC3-LAMP1 colocalization (Figure 9D), indicating the inhibition of autophagosome–lysosome fusion. Lysosome pH was also evaluated using Lyso-Tracker Red dye, which revealed a significant quenching in red fluorescence in cells treated with CPUL1 relative to the control cells (Figure 9E), presumably implying that CPUL1 induced a pH-dependent lysosomal dysfunction. Additionally, Western blotting analysis revealed that the expression of RAb7, LAMP1, and LAMP2, which are involved in lysosome membrane stability [35], were reduced by CPUL1 in a time-dependent manner (Figure 9F), suggesting the interference on lysosomal membrane function and obstruction in autophagy flux.

## 4. Discussion

Given its distressing prevalence and lethality, HCC has constituted and will likely continue to pose a severe threat to human health and life, necessitating the desperate development of effective chemotherapeutic agents [36]. Herein, in accordance with our previous reports, the therapeutic effects of CPUL1, a synthetic phenazine derivative, were systematically investigated against in vitro and in vivo HCC [10]. To elucidate the therapeutic mechanisms of this compound, comparative transcriptomics and untargeted metabolomics were performed for different exposure times (6 h or 48 h) in the HCC cell line BEL-7402. Multiple omics were combined to illustrate the biological processes dynamically and comprehensively within a genetic and phenotypic context, which interestingly underscored an aggravating metabolism dysfunction and nutritional deficiency. Thereupon, this molecule’s interception on autophagy flux was confirmed, which might provide unexpected insights into its underlying mechanisms and shed light on validating chemotherapeutic approaches against HCC in the context of metabolic intervention.

In vitro CPUL1 inhibited cell proliferation in different human-derived HCC cells (HepG2, BEL-7402, and HUH-7), as indicated by the low IC_50_ values (<10 μM). The spatiotemporal localization of CPUL1 was the first to reveal the predominant distribution in cytoplasm and mitochondria, accompanied by the gradual aggregations in mitochondria and nuclei, seemingly justified in part by the mitochondrial apoptosis and DNA damage observed in previous work [11]. The significant antineoplastic efficacy of the compound was confirmed in vivo on BEL-7402 xenografted nude mice, demonstrating an equivalent or even better effect than sorafenib or CTX (20 mg/kg) at the higher dosage (40 or 20 mg/kg). Given that the xenograft models were performed in immunocompromised animals, which eliminates the contribution of the host immune system, logically, the therapeutic effects of CPUL1 should attribute to its direct actions rather than immunological regulation.

Hence, the transcriptomic and metabolomic data from CPUL1 short- (6 h) or long-exposure (48 h) stages in BEL-7402 were analyzed to comprehensively understand the mechanisms against HCC. As for RNA profiling, the panoramic composition of DAGs apparently varied at different phases, however, the most drastic decreases coincided principally in several genes including COL12A1, FAT1, FBN2, IGF2R, and LRP1 (Figure 2A). Of note, recent reports have underscored the pivotal roles of these genes in tumorigenesis, invasion, and prognosis (e.g., COL12A1 in gastric cancer [37,38], FAT1 [39,40], and FBN1/2 [41,42] in HCC). Given the considerable degree and duration, the transcriptional suppressions on these genes may entail the disruption of respective neoplastic signals, fragmentarily elaborating the chemotherapeutic actions of CPUL1. Interestingly, the GO term or KEGG enrichment analysis showed that these DAGs from different time stages were mapped into a range of biological processes that only partially overlapped, supposedly indicating a temporal dissimilarity in the global transcriptional response. Further analysis of the interaction networks and hub genes demonstrated a time variation in transcriptional regulations, which brought about changes in the signaling processes (e.g., diverse cytokine pathways: EGFR, TGF-β, wnt), stress responses (e.g., DNA repair, ribosomal function, and RNA transport), metabolic processing (e.g., lipid metabolism, and glycan biosynthesis), ECM organization (e.g., ITGA, ITGB, LAMA, LAMB, and LAMC), endocytosis and protein recycling (e.g., collagen family), and cell cycle (Appendix A).

As an emerging approach, metabolomics could provide a direct and comprehensive assessment of phenotype through the functional readout of metabolic processes to various environmental or genetic changes, highlighting the potential to significantly impact the oncology and intervention core areas [43]. In the present study, metabolomics offered chronologic snapshots of phenotypic status concerning CPUL1-triggered changes, which could present a non-overlapping perspective to substantiate the consequence of transcriptional regulation. The overall composition of DAGs under 6 h- or 48 h-CPUL1 treatment was inconsistent, perceivably describing a discrepancy in metabolic status. A typical case was α-KG, a signaling metabolite in carbon metabolism, of which the variation pattern was only reversed at different dosing phases, with the FC values (vs. control) boosting from the early (FC ≈ 0.40) to the advanced stage (FC ≈ 3.51) in abundance. In the recent two decades, the crucial and multifactorial contribution of α-KG has hogged the limelight in oncogenesis and tumor suppression. As for fast-proliferating cells, α-KG, often derived from glutamine, tends to influx into the anaplerotic pathway to replenish the TCA cycle for macromolecules biosynthesis and energy production, which hence rewires cellular metabolism for oncogenesis and proliferation [44]. On the other hand, α-KG is also a pivotal regulator of hypoxic adaptations and epigenetic modifications, two of the most significant drivers of tumor transformation [45], which could even open a therapeutic window to impinge on tumor progression and metastasis [46]. Therefore, the variation in the α-KG abundance might represent the outcome of metabolic events and the dynamic adjustment of physiological status, as illustrated by the widely perturbed genes and metabolites.

Global functional annotations reflected an inactive scenario given the pervasive suppression of multiple metabolic pathways and corresponding metabolites, despite exceptional increments in several lipid metabolites (Figure 7). More specifically, the upregulated genes by CPUL1 exposure were significantly enriched within the pathway of unsaturated fatty acid (UFA) metabolism; correspondingly, a variety of metabolites including oleic acid (C18:1; FC ≈ 0.50/7.35, 6 h/48 h), linoleic acid (C18:2; FC ≈ 5.39/9.26, 6 h/48 h), α-linolenate (C18:3; FC ≈ 4.31/8.72, 6 h/48 h), and arachidonate (C20:4; FC ≈ 2.42/9.11, 6 h/48 h) were significantly enhanced, with an accentuating drift in abundance (Figure 5B,C). Increasing evidence has proposed the contribution of lipid remodeling to hepatic carcinogenesis and addressed the positive correlation between the content of lipid desaturation with malignant degrees and poor prognosis, which is potentially attributable to hypoxia adaptation, stress rescue, and immune regulations [47,48,49,50]. In our study, the UFAs were characterized to accumulate in the presence of CPUL1, which presumably meant the promotion of cellular proliferation according to the findings above, interestingly, contradicting its cytotoxic effects. Such a seeming paradox might be ascribed to the reduction in utilization due to CPUL1 treatment, which could nudge homeostatic lipid metabolism to cellular deposition given the steady and capped supply of these UFAs as a consequence of exogenous uptake, especially for essential fatty acids (e.g., linoleic and linolenic acids) [51]. In contrast, the excessive de novo biosynthesis of lipids, a hallmark of oncogenesis, possibly hindered by CPUL1, speculated from the consistent drops in the content of sn-glycerol-3-phosphate (FC ≈ 0.48/0.40, 6 h/48 h), CDP-choline (FC ≈ 0.96/0.29, 6 h/48 h), and octadecanoic acid (C18:0; FC ≈ 0.38/0.49, 6 h/48 h). Numerous metabolomic investigations have confirmed the correlations between the aberrant activation of lipogenesis and hepatocarcinogenesis and development, which could ensure tumor cells with extra lipids for membrane formation, biofuel supply, or post-translational modifications [48]. In this regard, a range of fatty acids have been explicated to abnormally accumulate in HCC such as octadecanoic acid (C18:0) [52,53], triglycerides (TG) [54], and phosphatidylcholine (PC) [55], which were significantly associated with the malignant degrees and thus identified as potential biomarkers for risk and prognostic assessment [56]. Coincidentally, sn-glycerol-3-phosphate and CDP-choline, whose concentrations decreased significantly according to our data, are the precursor and key intermediate metabolites for triglyceride generation [57] and PC biosynthesis [58], respectively, implying severe disturbances in these lipid anabolisms. Therefore, within the context of lipid utilization, it is plausible to infer that CPUL1 therapy may impede the availability and accessibility of intercellular lipids, potentially and profoundly altering a range of pathophysiological processes.

Moreover, the therapeutic efficacies of CPUL1 are also reflected in the reversal of the elevated metabolism of glucose, amino acids, and nucleotides, which are acknowledged to accommodate the enhanced demand for energy production and building blocks in hyperplastic cells [59]. For instance, as the exposure time extended, the abundance of several intermediate products in central carbohydrate metabolism (e.g., ribulose, D-erythrose 4-phosphate, 6-phospho-D-gluconate) remained low, resulting in dramatic content decreases of 3-phospho-D-glycerate (FC ≈ 0.20) and phosphoenolpyruvate (FC ≈ 0.15) after 48 h of treatment. In terms of the highly intertwined nature of metabolic networks, the recessions were evident in the pentose phosphate pathway, nucleotide synthesis, and amino acid metabolism, characterized by the continuous decreases in extensive end products such as adenine (FC ≈ 0.19), guanosine (FC ≈ 0.32), thymine (FC ≈ 0.35), ATP (FC ≈ 0.34), UDP (FC ≈ 0.26), CMP (FC ≈ 0.32), and glycine (FC ≈ 0.44), at 48 h of the administration of CPUL1. Of note, the relative content of γ-glutamylcysteine (GGC) and S-adenosyl-L-homocysteine (SAH), precursors to glutathione and cysteine, dropped substantially over the treated course, with FC values of 48 h approximately in 0.21 and 0.20, respectively. Accordingly, the accumulation of glutathione disulfide (GSSG; FC ≈ 2.67, 48 h) followed the rapid decline (FC ≈ 0.43, 6 h) in glutathione (GSH), apparently consistent with the oxidative damage induced by CPUL1 in previous reports, in light of the inclusive and evident deficiencies of GSH biosynthetic substrates [60]. Collectively, as far as transcriptional and metabolomic data are concerned, after CPUL1 treatment, the metabolic flux experienced a full range of decline (Figure 6 and Figure 7), which limited intracellular nutrition availability and potentially aggravated oxidative stress.

Given the scenario of nutritional deprivation induced by CPUL1, the supposed cytotoxic agent, it is obvious to question the potential participation of autophagy in the process, assuming its nature as a cytoprotective mechanism in reaction to starvation [61]. Hence, we further investigated the effect of CPUL1 on autophagy and found that LC3 conversion was markedly stimulated, with an accumulation of the LC3-II protein, the marker for autophagosome formation, which could technically be ascribed to either an increase in autophagosome formation or a decrease in autophagic turnover. Next, we assessed the expression of the p62 protein, a classical cargo adaptor for the degradation of ubiquitinated substrates via autophagy, which was time-dependently elevated, presumably indicating an obstruction in autophagic flux. Additionally, the observation of the comparable evidence of the autophagy flow inhibition in the HepG2 and HUH-7 cell lines, as evidenced in Figure 8F, implies a non-specific effect on the autophagy of CPUL1. Subsequent morphological and fluorescent profiling validated the above presumption, demonstrating a late-stage suppression of autophagy by CPUL1, especially in light of the discrepant effects induced by the co-treatment with different types of inhibitors (CQ or 3-MA). 

Interestingly, neither inhibitor, CQ or 3-MA alone, could induce significant cell apoptosis or suppress proliferation [29], indicating the cytotoxicity of CPUL1 could not be primarily ascribed to autophagy inhibition. Nevertheless, the suppression of autophagic flux should be addressed due to the comprehensive but consistent modulation of molecular events by the CPUL1 challenge, from genetic to protein to phenotypic levels. In the case of transcriptomics, LRP1, a functional gene implicated in intracellular signaling and endocytosis [62], was highlighted among those drastically downregulated genes, in contrast to other counterparts with direct entanglement in carcinogenesis and development (e.g., COL12A1, FBN1/2 [63], MET [64], LAMB1/2 [65], FAT1 [66], and IGF2R [67]). The gene product, a multifunctional and ubiquitous member of the LDLR family, low-density lipoprotein receptor-related protein 1 (LRP1), has been implicated in numerous pathophysiological processes including lipid and lipoprotein metabolism, protease degradation, and cell migration [68]. Previous observations have established that the LRP1 deficiency exacerbated hepatic susceptibility to palmitate-induced lipotoxicity in vitro and in vivo due to impairment in autophagic flux, as evidenced by the increased p62 levels but unaltered LC3 lipidation [69,70]. A similar scenario was described in the etiology and pathogenesis of developmental dysplasia of the hip, underlining a critical role of LRP1 in triradiate cartilage development via autophagy activation [71]. In this context, the substantial decline in the LRP1 transcript corresponded to the enrichment in the autophagy-related pathways in transcriptomics and the consequences of extensive metabolic abnormality and conceivable nutritional deprivation in metabolomics. Regarding our current approach, multi-omics data integration highlighted the significance of autophagic inhibition in the antineoplastic effects of CPUL1, which presumably conferred a nutrient deprivation by tapping into tumor-specific metabolic vulnerability and thus exacerbated its cytotoxicity. Indeed, the underlying mechanisms comprised of intricate, multivariate, and intertwined biochemical events would be an issue for future investigations.

Lysosomes are commonly considered as the primary location for intracellular degradation, where autophagosomes fuse with lysosomes to enable the degradation of their contents. Therefore, blocking lysosomal and autophagolysosomal activities significantly impairs autophagic flux [72]. Our results demonstrate that CPUL1 treatment disrupted lysosome–autophagosome fusion and resulted in the accumulation of immature forms (Figure 9D). Additionally, CPUL1 administration led to lysosomal dysfunction represented by lysosomal alkalization (Figure 9E) and the reduced expression of lysosomal membrane-associated proteins (LAMP1 and LAMP2) as well as the late endosomal/lysosomal regulator RAb7 (Figure 9F). Notably, CPUL1 had a comparable effect to CQ in terms of interference with the late stage of autophagy, inducing destructive autophagy and cytotoxic consequences [73].

Accumulating research has underscored the correlation between autophagy dysregulation and tumorigenesis and treatment regimens, which is proposed as a cytoprotective mechanism conferring resistance to stress including chemotherapy and radiation [74,75]. For instance, sorafenib, the first-line therapy for advanced HCC, resulted in tumor stabilization and cytostatic rather than regression. The relative resistance is presumably due to enhanced autophagy [76]. Hence, the strategies in combination with autophagy suppression have attracted significant interest in improving medication outcomes by restoring cellular sensitization to cancer therapeutics [77]. Therefore, the attractive properties of CPUL1, comprising of cytotoxicity and autophagic blockage, might endow the potential to translate this compound into a promising anti-HCC agent, necessitating additional studies.

## 5. Conclusions

The present study systemically demonstrated the therapeutic efficiency of CPUL1, a phenazine derivative, by inhibiting proliferation and inducing the apoptosis of HCC cells in vitro and in vivo. Subcellular localization suggested that CPUL1 accumulated in the cytoplasm and mitochondria, supposedly plotting the spatiotemporal context for previous observations that CPUL1 triggers mitochondrial apoptosis and ROS stress. Subsequent multi-omics analysis collectively profiled a scenario of time-dependent deterioration in metabolic dysfunction and nutritional depletion, which might be ascribed, at least partially, to the concurrent autophagy inhibition. Our results illustrate that CPUL1, a putative TrxR1 inhibitor, might mediate apparent metabolic stress by inhibiting cellular autophagy, in addition to the anticipated oxidative stress and subsequent cellular lesion. This multifaceted mechanism of action endows the compound with intriguing biological properties to sensitize malignant cells to damage, and highlights its potential promise as a therapeutic agent against HCC as well as providing insights into the development of innovative strategies for combating cancer. 

## Figures and Tables

**Figure 1 cancers-15-01607-f001:**
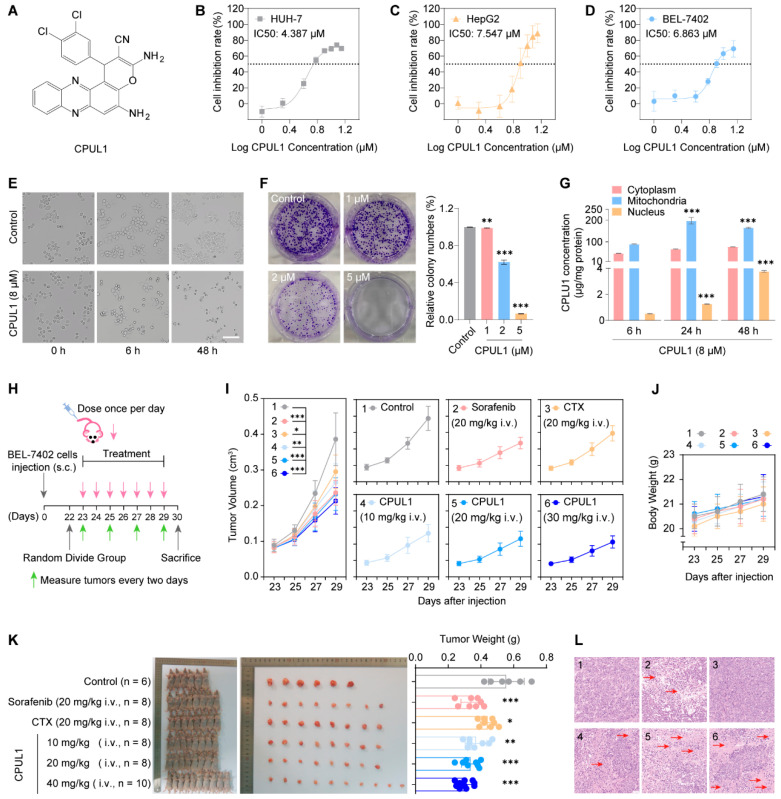
The structure of CPUL1 and the effects of CPUL1 on the viability of liver cancer cells in vitro and the antitumor effects in vivo. (**A**) The structure of CPUL1. (**B**–**D**) Liver cancer cell lines were treated with CPUL1 (0–14 μM) for 48 h, and the CCK-8 assay measured cell viability. The absorbance was measured at 450 nm. (**E**) The cell morphology of BEL-7402 cells after 48 h of continuous treatment with CPUL1 (8 μM), and the number of cells was significantly reduced. (**F**) The cells were treated with CPUL1 (8 μM) for 14 days, and the cell colony formation assay determined the clone-forming ability. The histograms showed the quantified results of the colony formation assay. (**G**) The sub-organellar localizations of CPUL1 in the BEL-7402 cells. The concentration of CPUL1 in the mitochondria and nucleus at 24 h and 48 h was compared with the time point of 6 h, respectively. (**H**) Schematic of the animal model establishment, treatment, and measurement. (**I**,**J**) The tumor volume and body weight was measured every two days. (**K**) The xenograft tumors were separated when the animals were killed. The final tumor weights were measured. (**L**) Histopathological tests (hematoxylin and eosin staining) were performed on the BEL-7402 transplanted tumor tissues. Red arrows indicate necrotic areas. Scale bar = 50 μm. The data were expressed as the mean ± SD; * *p* < 0.05, ** *p* < 0.01, *** *p* < 0.001. ns, not significant.

**Figure 2 cancers-15-01607-f002:**
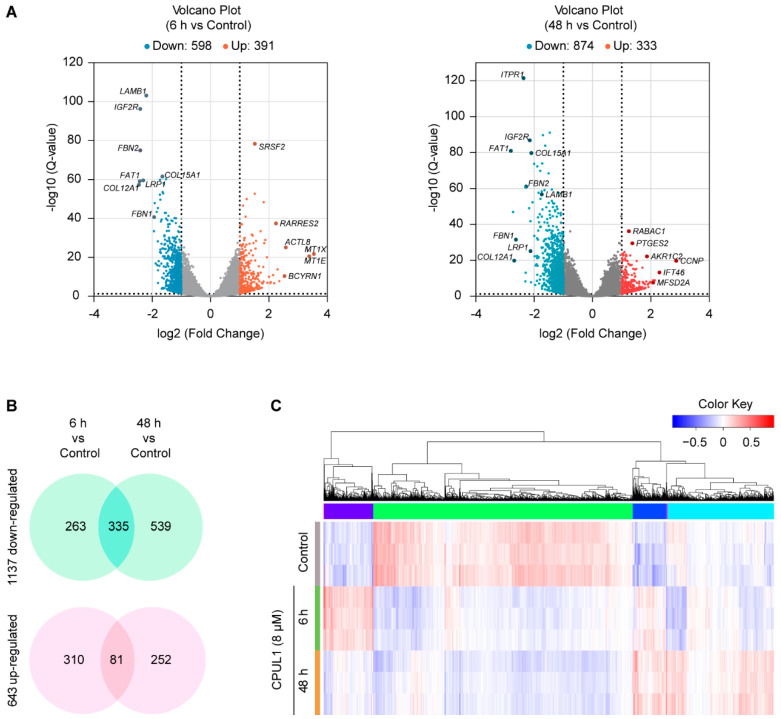
Transcriptome analysis of BEL-7402 exposed to CPUL1. (**A**) Volcano diagram of the DEGs between 6 h- and 48 h-CPUL1-treated BEL-7402 cells and control cells (n = 3). The y-axis corresponds to the expression value of −log10 (*Q*-value), and the x-axis displays the log2 (FC) value. Red dots indicate upregulated genes, blue dots indicate downregulated genes, and black ones represent the genes without statistical significance. (**B**) Venn diagram of DEGs. The numbers of DEGs are indicated. (**C**) The cluster heatmap of upregulated and downregulated genes in the 6 h- and 48 h-CPUL1-treated BEL-7402 and control cells.

**Figure 3 cancers-15-01607-f003:**
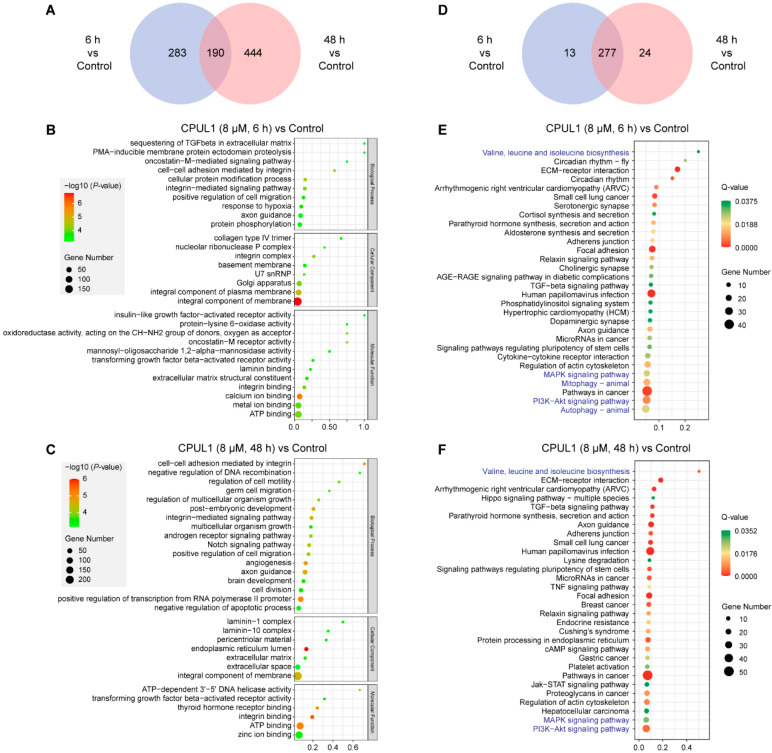
Top 30 enriched Gene Ontology (GO) terms and Kyoto Encyclopedia of Genes and Genomes (KEGG) pathway analysis on DEGs. (**A**) Venn diagram of GO terms. The numbers of GO terms were indicated. (**B**) GO terms of the cell’s DEGs under CPUL1 exposure of 6 h. (**C**) The GO terms of the DEGs of the cell under CPUL1 exposure for 48 h. The vertical and horizontal axes represent the GO terms and −log10 (*p*-value) for each term, respectively. (**D**) Venn diagram of KEGG classifications. The number of KEGG classifications is indicated. (**E**) Bubble chart of the KEGG classifications of assembled DEGs under CPUL1 exposure for 6 h. (**F**) Bubble chart of the KEGG classifications of the assembled DEGs under CPUL1 exposure for 48 h. The vertical axis represents the KEGG terms. The size of the circles roughly represent the count of annotated genes. The color saturation from green to red indicates the *Q*-value.

**Figure 4 cancers-15-01607-f004:**
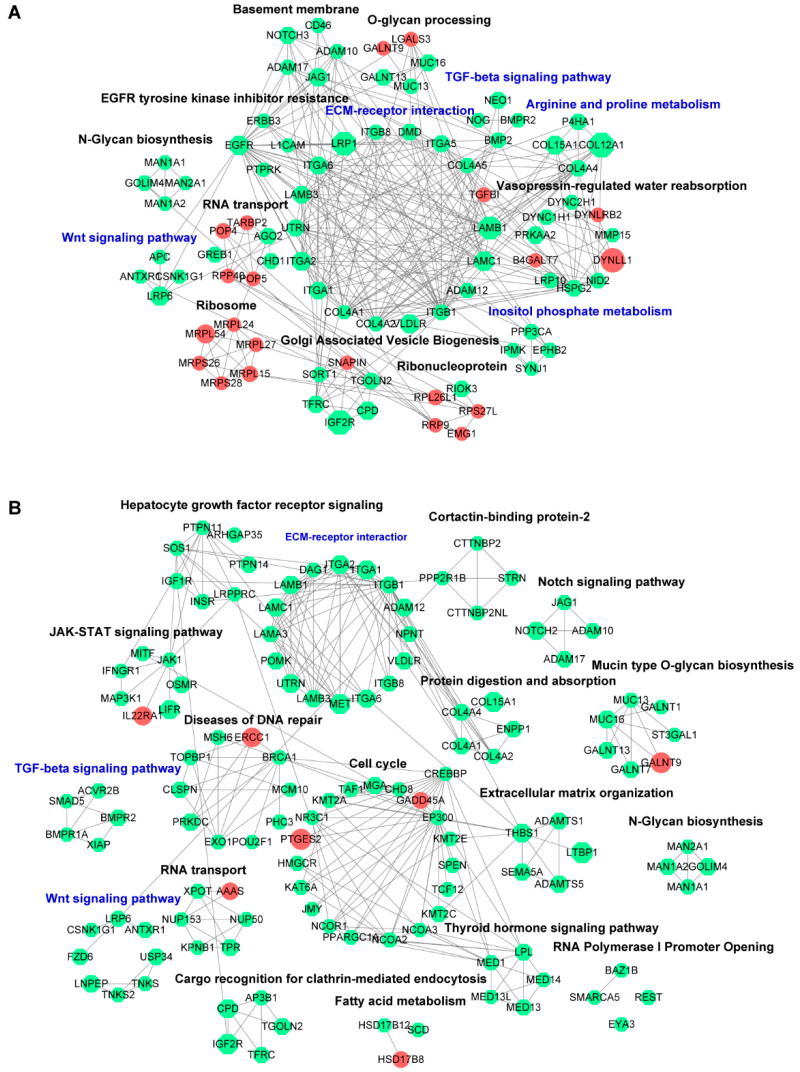
Protein–protein interaction network analysis. (**A**) Interactions of upregulated and downregulated differentially expressed genes (DEGs) under CPUL1 exposure for 6 h. (**B**) Interactions of upregulated and downregulated DEGs under CPUL1 exposure for 48 h. The predicted protein products of DEGs were entered in the STRING database (http://string-db.org/ accessed on 1 November 2022), setting an interaction score of 0.900. The nodes represent the protein products of the DEGs. Red and green represent the upregulation and downregulation, respectively, with the size correlative to the log2 (FC) value. Edges indicate the functional connections among nodes. The putative categories were annotated based on the functional association. DEGs were differentially expressed genes.

**Figure 5 cancers-15-01607-f005:**
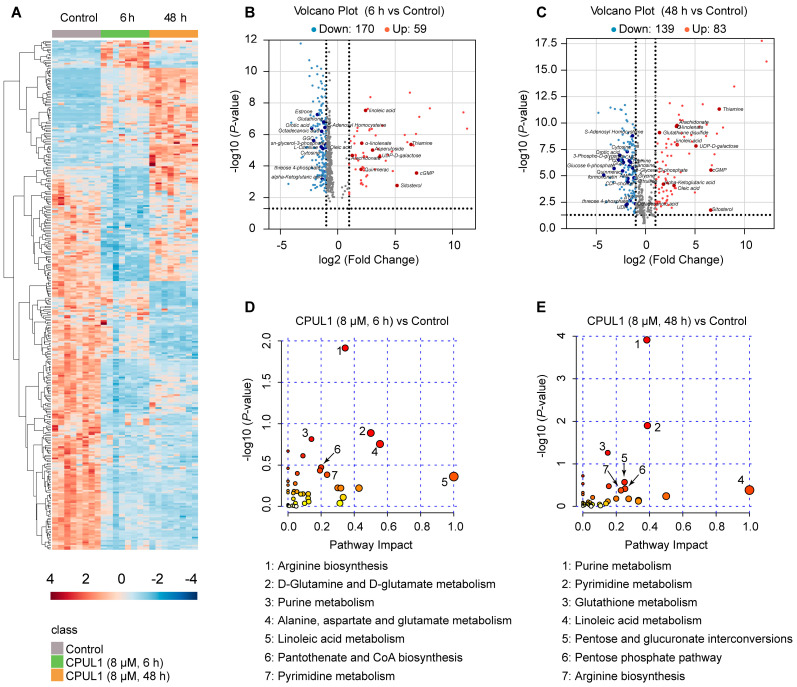
The effects of CPUL1 on the metabolic pathways of the BEL-7402 cells pretreated with 8 μM CPUL1 for 6 h and 48 h. (**A**) The cluster heatmap of the most significantly differential abundance metabolites (DAMs) identified for CPUL1 treatments and control, and the blue to the red color scale indicated the value of DAM expression after the scale number. (**B**,**C**) Volcano diagram of the DAMs between 6 h- and 48 h-CPUL1-treated BEL-7402 cells and control cells (n = 8). The y-axis corresponds to the expression value of −log10 (*p*-value), and the *x*-axis displays the log2 (FC) value. Red dots indicate upregulated genes, blue dots indicate downregulated genes, and black ones represent the genes without statistical significance. (**D**) Metabolome view of the pathway analysis generated using MetaboAnalyst 5.0 based on DAMs in the untreated and 6 h-CPUL1-treated cells. (**E**) Metabolome view of the pathway analysis generated using MetaboAnalyst 5.0 based on DAMs in the untreated and 48 h-CPUL1-treated cells.

**Figure 6 cancers-15-01607-f006:**
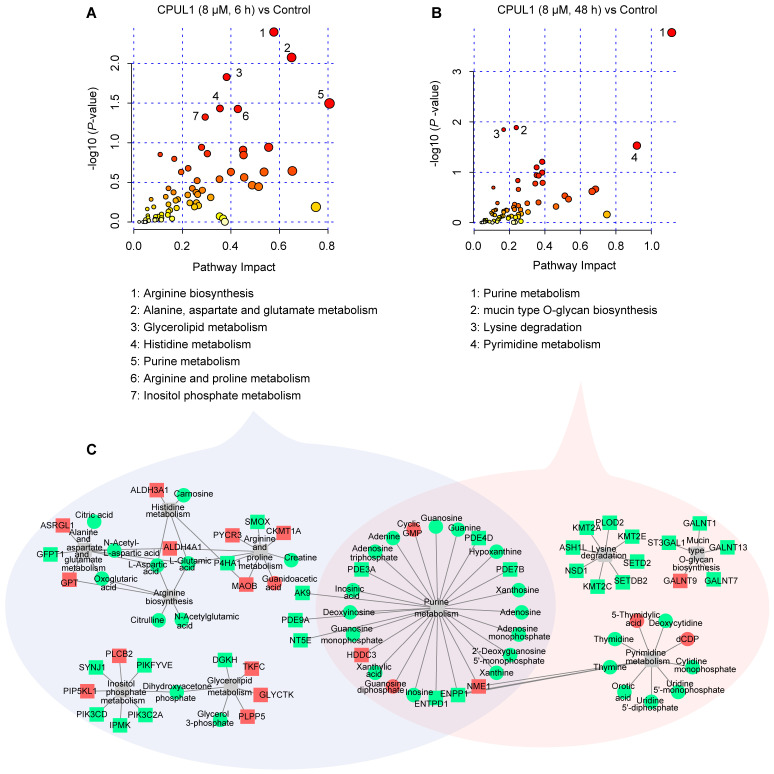
Integrated analysis of the mechanism underlying the effects of CPUL1 treatment on BEL-7402 cells based on the metabolomic and transcriptomic data. Pathway analysis was generated using MetaboAnalyst 5.0 based on differential genes and metabolites. (**A**) The 6 h-CPUL1-treated group. (**B**) The 48 h-CPUL1-treated group. (**C**) Visualization of the pathways comprised of genes and metabolites. The square dots represent genes, and the round dots represent metabolites. Red and green represent the upregulation and downregulation, respectively. Edges represent the functional connections between the pathways and genes or metabolites.

**Figure 7 cancers-15-01607-f007:**
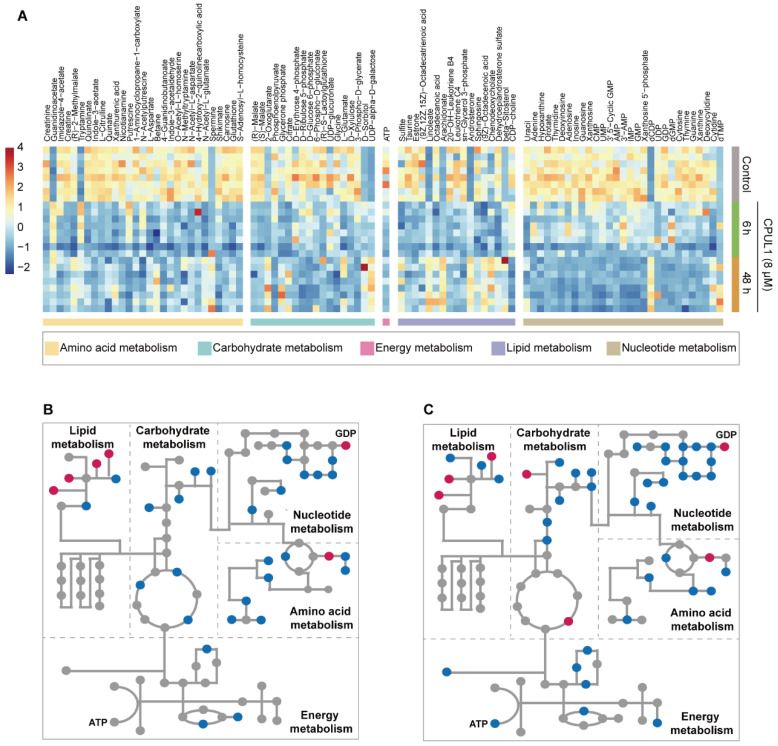
Differentially expressed endogenous metabolites enriched in the pathways. (**A**) The cluster heatmap of material metabolism-related metabolites identified for CPUL1 treatments and the control and the blue to the red color scale indicates the value of DAM expression. (**B**) Compounds with endogenous differences in the lipid, carbohydrate, nucleotide, amino acid, and energy metabolism under the exposure of 6 h-CPUL1-treatment. (**C**) Compounds with endogenous differences in the lipid, carbohydrate, nucleotide, amino acid, and energy metabolism under the exposure of the 48 h-CPUL1-treatment.

**Figure 8 cancers-15-01607-f008:**
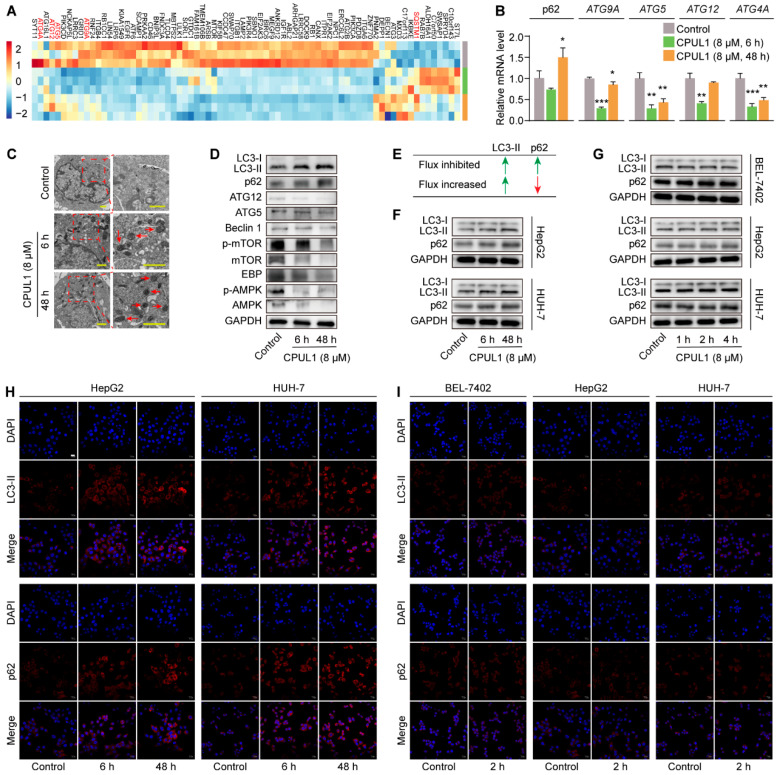
Effects of CPUL1 on autophagy pathway. (**A**) The cluster heatmap of autophagy-related genes was identified for CPUL1 treatments and the control, and the blue to the red color scale indicate the value of the differentially expressed gene (DEG) expression. (**B**) QPCR analysis of p62, ATG9A, ATG5, ATG12, and ATG4A in the control and CPUL1-treated BEL-7402 cells. Cells were treated with CPUL1 (8 μM) for 6 h and 48 h. (**C**) The cells were treated with or without CPUL1 (8 μM) for 6 h and 48 h. Autophagosomes were observed by TEM. Scale bar = 1 μm. (**D**) Western blot analysis of LC3, p62, ATG12, ATG5, Beclin1, mTOR, p-mTOR, EBP, AMPK, and p-AMPK in the control and CPUL1-treated BEL-7402 cells. Cells were treated with CPUL1 (8 μM) for 6 h and 48 h. (**E**) Schematic representation of the LC3-II and p62 levels under increased or inhibited autophagy flux. Green arrow indicated increase and red arrow indicated decrease. (**F**) HepG2 and HUH-7 cells were treated with 8 μM CPUL1 for 6 h or 48 h, and the levels of LC3- II/ I and p62 were assessed by western blotting. (**G**) BEL-7402, HepG2, and HUH-7 cells were treated with 8 μM CPUL1 for 1 h, 2 h, and 4 h, respectively, and the level of LC3- II/ I and p62 were analyzed by Western blotting. (**H**) HepG2 and HUH-7 cells were treated with 8 μM CPUL1 for 6 h and 48 h, and immunostained with p62 and LC3 antibodies. DAPI (blue) was used to stain the nuclei, and the fluorescent images were observed with a confocal laser scanning microscope (Olympus FV3000). Scale bar = 20 μm. (**I**) BEL-7402, HepG2, and HUH-7 cells were treated with 8 μM CPUL1 for 2 h, and immunofluorescence labeling with p62 and LC3 antibodies was performed. DAPI (blue) was used to stain the nuclei, and the fluorescent images were observed with confocal laser scanning microscopy (Olympus FV3000). Scale bar = 20 μm. The data were expressed as the mean ± SD; * *p* < 0.05, ** *p* < 0.01, *** *p* < 0.001. ns, not significant.

**Figure 9 cancers-15-01607-f009:**
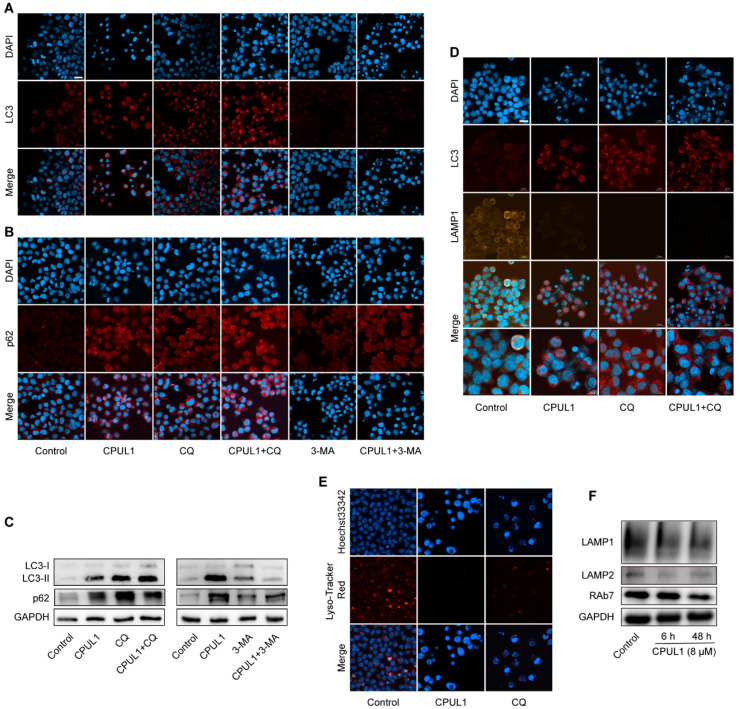
CPUL1 interferes with lysosomal function and fusion with autophagosomes. (**A**,**B**) The cells were treated with CPUL1 (8 μM) and 3-MA (2 mM) or CQ (20 μM) for 48 h, immunolabeling with the p62 and LC3 antibodies. DAPI (blue) was used to stain the nuclei, and the fluorescent images were observed with laser scanning confocal microscopy and processed using the ZEN imaging software. Scale bar = 20 μm. (**C**) The cells were treated with CPUL1 (8 μM) and 3-MA (2 mM) or CQ (20 μM) for 48 h, and the level of LC3-II/I and p62 was detected by Western blotting. (**D**) The cells were treated with CPUL1 (8 μM) and CQ (20 μM) for 48 h, and the colocalization of LC3 (667 nm red) and LAMP1 (570 nm orange) was assessed. DAPI (blue) was used to stain the nuclei, and the cells were photographed under a fluorescence microscope. Scale bar = 20 μm. (**E**) Cells were treated with CPUL1 (8 μM) and CQ (20 μM) for 48 h and stained with Lyso-Tracker Red for 60 min, Hoechst 33342 (blue) was used to stain the nuclei and visualized under a fluorescence microscope. Scale bar = 20 μm. (**F**) Cells were treated with CPUL1 (8 μM) for 6 h and 48 h, and the levels of RAb7, LAMP1, and LAMP2 were detected by Western blotting.

## Data Availability

The data presented in this study are available in Appendix A.

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
