# Peer review of "A Novel Phenazine Analog, CPUL1, Suppresses Autophagic Flux and Proliferation in Hepatocellular Carcinoma: Insight from Integrated Transcriptomic and Metabolomic Analysis"

_cancers, 2023, doi:10.3390/cancers15051607_

Round 1

Reviewer 1 Report (Previous Reviewer 2)

The authors successfully addressed all my comments.

Reviewer 2 Report (Previous Reviewer 1)

The authors addressed the questions asked. 

This manuscript is a resubmission of an earlier submission. The following is a list of the peer review reports and author responses from that submission.

Round 1

Reviewer 1 Report

The authors aim to understand the mechanism of the anti tumorigenesis properties of CPUL in HCC. For that purpose they undertake a variety of approaches, both in vivo and in vitro, as well as several large scale studies for better describe the effects of this drug in CPUL. Although it is a nice descriptive piece of work there is very little mechanism provided. Regardeless of this point, there are a few important questions that should be addressed to be able to fully support the conclusions raised.

Figure 1: The different treatments in the mice have some effect on tumor growth, but it would be very interesting to see what are the effects of combination treatments, such as CPUL+Sorafenib, and CPUL+CTX. I would suggest to do these experiments both in vitro in the three cell lines in 1B, as well as in the xenograft model. In addition, it would add a lot the the first figure showning the data in E, F, G also in the other two cell lines, as in B.

As for the tumor xenografts, H&E of at least of some representative tumors should be presented to see if they recapitulate the morphology changes seen in the cells in culture. In addition, the effects on signaling pathways shown at the end of the manuscript, such as mTOR (phospho S6 (S240/244) for example), AMPK (phospho-ACC), and autophagy markers (p62 levels) should be analyzed in the tumors, either by WB or by IHC.

Figure 2: A validation of some of the genes of the transcriptome analysis by RTPCR in cell lines and in xenografts should be done. It would be particularly interesting for p62 since it is transcriptional regulated.

Figure 8: Some of the WB analysis reflect a change in the total protein levels, then the phosphorylation status of that protein is hard to interpret. In addition, instead of phospho-mTOR or phospho-AMPK a better readout might be its downstream targets such as S6p, S6Kp, 4EBP1 or ACCp.

In figure 8A it would be interesting to see what are the levels of genes directly on the autophagy pathway rather than regulatory genes that affect the pathway, as well as a validation of ATG genes by RTPCR.

LC3 should be quantified using some microscopy method by number of dots per cell in at least 50 cells per condition in triplicate, and always adding bafilomycin A1 or CQ as controls.

In figure 8G the WB results seem quite different when comparing UT and CPUL in both pannels, and in one there is not much effect whereas in the other one an increase in p62 levels is observed. Why?

If CPUL inhibits autophagy, is this important for the effects seen on colony formation and cell growth or it is just a secondary response on the cells? Would an autophagy inhibitor have the same effect as CPUL in tumor formation, colony formation, cell growth… At least in vitro experiments should be performed towards this direction.

Reviewer 2 Report

The manuscript by Chen and Ge et al. reports CPUL1 as a new therapeutic target in hepatocellular carcinoma (HCC). The authors demonstrate that CPUL1 has antitumoral effects in vitro and in vivo in human HCC cells. They identified by transcriptomics and metabolomics analysis that CPUL1 treatment induces metabolic dysregulation and deprivation of nutrients in HCC cells. They describe that the autophagic flux degradation is impaired upon CPUL1 treatment in HCC as a potential mechanism involved in the metabolic perturbation induced by CPUL1. Overall, the data are well-presented, convincing, and supporting the conclusions. The findings provide new insights into the potential therapeutic role of CPUL1 in human HCCs. I have comments for improvement as outlined below, but provided these points are sufficiently addressed, I would be supportive of publication. The comments are not listed in any order of priority:

-       The autophagic flux is a rapid mechanism with a fast turnover, especially in starvation-like conditions such as CPUL1 treatment (low nutrients). Here, the authors described the impairment of the autophagic flux at 48 hours (Figure 8B, 8E-G). Do the authors detect similar effects on autophagic flux (LC3B-II and p62 increase) at shorter time points (either 1, 2 or 4 hours) upon CPUL1 treatment by immunoblotting and immunofluorescence?

-       The authors identified the role of CPUL1 in blocking the degradation of autophagosomes (Figure 8). As lysosomes are responsible for this degradation, do the authors observe any defects in lysosome morphology, numbers and/or function upon CPUL1 treatment?

-       Do the authors notice similar results on the impairment of the autophagic flux degradation upon CPUL1 treatment in other HCC cell lines (Huh7 and/or HepG2)?

-       Immunoblotting pictures in Figure 8B and 8G (left) does not reflect a significant increase in p62 levels upon CPUL1 (compared the Figure 8G right). Do the authors have other immunoblotting pictures supporting the findings?

-       The authors used “autophagy flow” and “autophagic flux” in their manuscript. For consistency in the manuscript and the literature, the authors should refer to this process using “autophagic flux”.

-       Could the authors specify the antibodies and dilution used for LC3 and p62 in their immunofluorescence experiments?

Reviewer 3 Report

In this study, authors used a multi-omics approach to elucidate the mechanism underlying the therapeutic efficacy of CPUL1, highlighting its involvement on autophagy dysregulation. Overall, the manuscript is well written. The introduction provides sufficient information to get the reader familiarised with the subject. The methods are very detailed and well described, although I would like to have more details on the statistical analysis and how the omics data were integrated statistically. Results are clear and appropriately described according to the aim. Perhaps it would be better to have the title of each sub-section indicating the main finding, instead of just a description of what was done, which we see already in the methods section. Figures are difficult to read and understand. Results were fairly discussed in the discussion, although it seemed to me that the transcriptomic data were not very explored. Therefore, given such minor suggestions, I recommend the publication of the manuscript in the present form.

Round 2

Reviewer 1 Report

Thanks for submitting this reviewed version of the manuscript. Although most of the questions were answered by providing new data or an explanation to it, there are a few points that were discussed on the previous version that are not properly adressed in this version:

1. The observed differences in the phosphorylated mTOR/AMPK are possibly due to changes in the total protein levels are observed in the western blot. That is ok but it must be carefully discussed in the manuscript.

2. The quantification of autophagosome formation provided into response 6 is not accurate. It was suggested to do number of dots rather than fluorescence intensity. The intensity of the fluorescence may have nothing to do with the autophagy status since that method does not provide information on where LC3 is located, at the cytosol or at the autophagosome. 

Reviewer 2 Report

The authors did not sufficiently addressed my comments. Although this is an interesting study, additional experiments are needed in other human hepatocellular carcinoma lines to validate their findings as suggested by other reviewers. The mechanism on how CPUL1 suppresses the autophagic flux is unclear and missing in the revised manuscript.